# Obesity Affects HDL Metabolism, Composition and Subclass Distribution

**DOI:** 10.3390/biomedicines9030242

**Published:** 2021-02-27

**Authors:** Julia T. Stadler, Sonja Lackner, Sabrina Mörkl, Athina Trakaki, Hubert Scharnagl, Andrea Borenich, Willibald Wonisch, Harald Mangge, Sieglinde Zelzer, Nathalie Meier-Allard, Sandra J. Holasek, Gunther Marsche

**Affiliations:** 1Division of Pharmacology, Otto Loewi Research Center, Medical University of Graz, Universitätsplatz 4, 8010 Graz, Austria; julia.stadler@medunigraz.at (J.T.S.); athina.trakaki@medunigraz.at (A.T.); 2Division of Immunology and Pathophysiology, Otto Loewi Research Center, Medical University of Graz, Heinrichstraße 31a, 8010 Graz, Austria; sonja.lackner@medunigraz.at (S.L.); nathalie.allard@medunigraz.at (N.M.-A.); 3Department of Psychiatry and Psychotherapeutic Medicine, Medical University of Graz, Auenbruggerplatz 31, 8036 Graz, Austria; sabrina.moerkl@medunigraz.at; 4Clinical Institute of Medical and Chemical Laboratory Diagnostics, Medical University of Graz, Auenbruggerplatz 15, 8036 Graz, Austria; hubert.scharnagl@medunigraz.at (H.S.); harald.mangge@medunigraz.at (H.M.); sieglinde.zelzer@medunigraz.at (S.Z.); 5Institute for Medical Informatics, Statistics and Documentation, Medical University of Graz, Auenbruggerplatz 2, 8036 Graz, Austria; andrea.borenich@medunigraz.at; 6Division of Physiological Chemistry, Otto Loewi Research Center, Medical University of Graz, Neue Stiftingtalstraße 6, 8010 Graz, Austria; willibald.wonisch@medunigraz.at; 7BioTechMed Graz, Mozartgasse 12/II, 8010 Graz, Austria

**Keywords:** obesity, HDL-C, HDL subclasses, LCAT, CETP, adiponectin, soluble leptin receptor

## Abstract

Background: Obesity increases the risk of coronary heart disease, partly due to its strong association with atherogenic dyslipidemia, characterized by high triglycerides and low high-density lipoprotein (HDL) cholesterol levels. Functional impairment of HDL may contribute to the increased cardiovascular mortality, but the effect of obesity on composition, structure, and function of HDL is not well understood. Design and Methods: We determined HDL composition, HDL subclass distribution, parameters of HDL function, and activities of most important enzymes involved in lipoprotein remodeling, including lecithin–cholesterol acyltransferase (LCAT) and cholesteryl ester transfer protein (CETP) in relatively young normal weight (*n* = 26), overweight (*n* = 22), and obese (*n* = 20) women. Results: Obesity (body mass index (BMI) ≥ 30) was associated with noticeable changes in LCAT and CETP activities and altered HDL composition, such as decreased apolipoprotein A-I, cholesterol, and phospholipid content, while pro-inflammatory HDL serum amyloid a content was increased. We observed a marked shift towards smaller HDL subclasses in obesity linked to lower anti-oxidative capacity of serum. LCAT activity, HDL subclass distribution, and HDL-cholesterol were associated with soluble leptin receptor, adiponectin, and liver enzyme activities. Of note, most of these alterations were only seen in obese women but not in overweight women. Conclusions: Obesity markedly affects HDL metabolism, composition, and subclass distribution linked to changes in liver and adipose tissue. HDL dysfunction may contribute to increased cardiovascular risk in obesity.

## 1. Introduction

One of the strongest predictors of cardiovascular disease in obesity and obesity-associated metabolic syndrome is a low level of high-density lipoprotein cholesterol (HDL-C) and increased triglyceride-rich lipoproteins (TRGLs) [1], which are components of a constellation, often referred to as “atherogenic dyslipidemia”. As a consequence of dietary changes and reduced physical activity, the increasing prevalence of obesity is becoming a severe burden for general public health [2]. Several diseases are known to be closely linked to obesity, such as cardiovascular disease, non-alcoholic fatty liver disease, cancer, and type 2 diabetes [3,4,5,6]. Reduced plasma levels of HDL-C, but increased triglyceride levels are often observed in obese individuals [7], together with an imbalance of adipokines, such as leptin and adiponectin [8]. Both peptide hormones are produced by adipose tissue and play an important role in the regulation of energy metabolism [9]. In obese individuals, adiponectin levels are decreased and associate with decreased HDL-C and HDL cholesterol efflux capacity, an integrated metric of HDL quantity and quality [10,11]. Interestingly, the most efficient treatment for low HDL-C levels in morbidly obese patients is bariatric surgery [12,13]. This surgical procedure leads to a remarkable increase in plasma HDL-C levels, but also to improved HDL function, independent of body weight [11,14,15].

Reduced HDL function may play a role in a variety of diseases, as HDL particles have anti-inflammatory [16], anti-oxidant [17], and anti-thrombotic [18] activities and promote reverse cholesterol transport [19]. HDL particles contain a considerable number of proteins and lipids and are structurally and functionally heterogeneous. Apolipoprotein (apo)A-I and apoA-II, which stabilize HDL, as well as apoC-II and apoC-III, act as cofactors for enzymes involved in lipid metabolism [20,21]. Furthermore, several enzymes are associated with HDL, such as paraoxonase, which is a hydrolytic enzyme with a wide range of substrates and is partly responsible for the anti-oxidant and anti-inflammatory properties of HDL [22]. Lecithin–cholesterol acyltransferase (LCAT) and cholesteryl ester transfer protein (CETP) are key enzymes involved in the remodeling of HDL. By esterifying the free cholesterol of HDL, LCAT increases the HDL particle size and provides substrates for CETP [23]. Disturbances in the activity of these enzymes have been described in insulin resistance, type 2 diabetes, and liver disease [24,25]. Effects of obesity on HDL metabolism and function remain unclear.

In this cross-sectional study in relatively young and healthy women, we assessed whether obesity affects HDL metabolism, composition, and subclass distribution as well as metrics of HDL function. 

## 2. Materials and Methods

### 2.1. Recruitment and Group Characteristics

A total of 68 participants were enrolled: 26 normal weight, 22 overweight, and 20 obese women aged 18–39 years. All participants gave their written informed consent. The study population was a subgroup of a larger cross-sectional study (5 groups of different energy status, *n* = 107) [13,26,27] and was conducted according to the Declaration of Helsinki and was approved by the ethics committee of the Medical University of Graz on 2 June 2014 (MUG-26-383ex13/14). The study population was enrolled according to the following inclusion criteria: female, aged between 18 and 40 years. The participants were assigned to the groups according to the WHO-recommended body mass index (BMI) categories for relative weight classification for normal weight (18.5–24.9 kg/m²), overweight (25.0–29.9 kg/m²), and obesity (≥30 kg/m²) [28]. We included only metabolically healthy women with the following exclusion criteria: acute or chronic illnesses or infections, alcohol or drug abuse, statin medication, severe cognitive deficits, history of digestive tract diseases, history of gastrointestinal surgery, treatment with antibiotics and taking prebiotics or probiotics within the last 2 months, pregnancy or breastfeeding.

### 2.2. Body Mass Index (BMI) and Body Fat Measurement

The BMI (body weight [kg]/body height [m]²) was calculated and used for group allocation in accordance with the WHO categories. To obtain information on the fat distribution of participants, we measured the thickness of subcutaneous adipose tissue (SAT) at 8 clearly defined body sites by the standardized ultrasound method [29]. The sum of these 8 digits was calculated (D_INCL_), which is a reliable and representative measure of body fat percentage, even in obese people [30].

### 2.3. Plasma Lipids

Plasma lipids such as total cholesterol, triglycerides, and HDL-C were measured by enzymatic photometric transmission measurement (Roche Diagnostics, Mannheim, Germany). The concentrations of LDL-cholesterol were calculated by the Friedewald’s formula [31].

### 2.4. VLDL 

Plasma levels of very low density lipoprotein (VLDL) were measured using the Lipoprint System (Quantimetrix Corp., Redondo Beach, CA, USA), according to the manufacturer’s instructions. Serum was loaded on gel tubes and mixed with 200 µL of Lipoprint loading gel, containing a lipophilic dye, which binds proportionally to the lipids in the sample. Photopolymerisation was carried out for 30 min. Electrophoresis was performed for 60 min at 3 mA per gel tube and at a maximum delivery of 500 V. After a rest period of 30 min, gel tubes were scanned and analyzed using the Lipoware Software (Lipoware HDL Research LW03-v.16-134). 

### 2.5. Markers of Inflammation 

Markers of inflammation, namely, C-reactive protein and interleukin-6, were analyzed by a particle-enhanced turbidimetric assay and an electrochemiluminescent immunoassay (ECLIA), respectively, on a Cobas 6000 chemical routine analyzer (Roche Diagnostics, Mannheim, Germany).

### 2.6. ApoB-Depleted Serum

ApoB-depleted serum was prepared by the addition of 40 µL polyethylenglycol (Sigma-Aldrich, Darmstadt, Germany) (20% in 200 mmol/L glycine buffer) to 100 µL serum followed by gentle mixing [32]. Serum samples were incubated at room temperature for 20 min, and after centrifugation at 10,000 rpm for 30 min at 4 °C, the supernatant was collected, and samples were stored at −70 °C until usage.

### 2.7. HDL-Associated Proteins and Lipids

HDL-associated apoA-I, apoA-II, apoC-II, apoC-III, and apoE were determined by immunoturbidimetry [33]. Lipids including cholesterol, phospholipids, and triglycerides were determined using enzymatic methods, as previously described [34]. Cholesteryl ester was calculated as the difference between total cholesterol and free cholesterol, measured in apoB-depleted serum. All lipoprotein analyses were performed on an Olympus AU680 analyzer (Beckman Coulter, Brea, CA, USA), as previously described [33]. Serum amyloid A (SAA) was quantified using a commercially available kit (Invitrogen, Carlsbad, CA, USA), according to the manufacturer’s instructions.

### 2.8. HDL Particle Size

ApoB-depleted serum (2 µL) was separated by native gradient gel electrophoresis (4–16% NativePage; Life Technologies, Carlsbad, CA, USA). The gels were run for 120 min at constant voltage of 150 in NativePage running buffer (Life Technologies, Carlsbad, CA, USA). Subsequently, standard (NativeMark, Life Technologies, Carlsbad, CA, USA) of the gels was fixed with 25% isopropanol/10% acetic acid for 10 min and stained with protein staining solution (PageBlue, Thermo Scientific, Waltham, MA, USA). Separated neutral lipids of the samples were stained with Sudan black (Sigma-Aldrich, Darmstadt, Germany). Size distribution of HDL was analyzed using Image Lab software (version 5.2), as described previously [35].

### 2.9. Arylesterase Activity of Paraoxonase

Arylesterase activity of HDL-associated paraoxonase was determined with a photometric assay using phenylacetate as substrate, as described elsewhere [36].

### 2.10. Anti-Oxidative Capacity of HDL

The anti-oxidative activity of serum was determined with a fluorometric assay, as previously described [37]. The ability of serum samples to inhibit oxidation of dihydrorhodamine was monitored.

### 2.11. Total Oxidative Capacity (TOC) 

TOC captures the total peroxide concentration in serum. TOC was determined by a rapid enzymatic in vitro diagnostic assay (TOC, Labor Diagnostic Nord, Nordhorn, Germany). The assay uses a peroxide/peroxidase reaction with 3,5,3′,5′-tetramethylbenzidine as substrate. The results were calculated from the linear hydrogen peroxide standard curve and peroxide levels were specified as micromole H_2_O_2_ equivalents [38]. 

### 2.12. LCAT Activity

LCAT activity of serum was assessed by a commercially available kit (Merck, Darmstadt, Germany) according to the manufacturer’s instructions. Specifically, serum samples were incubated with the LCAT substrate for 4 h at 37 °C. The fluorescent substrate emits fluorescence at 470 nm. When the substrate is hydrolyzed by LCAT, a monomer is released that emits fluorescence at 390 nm. The LCAT activity is assessed over time and expressed in change of 470/390 nm emission intensity.

### 2.13. LCAT Protein Concentration

LCAT protein concentration in serum samples was measured using a commercially available ELISA kit (BioVendor, Brno, Czech Republic) according to the manufacturer’s instructions. 

### 2.14. CETP Activity

CETP activity of serum was measured using a commercially available kit (Abcam, Cambridge Science Park, Cambridge, UK), according to the manufacturer’s instructions. Specifically, the assay uses a donor molecule containing a fluorescent self-quenched neutral lipid that is transferred to an acceptor molecule in the presence of CETP. The CETP-mediated transfer of the fluorescent lipid to the acceptor molecule results in an increase in fluorescence intensity (excitation: 465 nm; emission: 535 nm).

### 2.15. Cholesterol Efflux Capacity of ApoB-Depleted Serum

Cholesterol efflux capacity was performed as described elsewhere [19,39]. J774.2 cells (Sigma-Aldrich, Darmstadt, Germany) were cultured in Dulbecco’s modified Eagle’s medium (Life Technologies, Carlsbad, California, USA) in the presence of 10% fetal bovine serum and 1% penicillin/streptomycin. A total of 300,000 cells per well were plated on 48-well plates (Greiner Bio-One, Kremsmünster, Austria), cultured for 24 h, and labelled with 0.5 µCi/mL radiolabeled [^3^H]-cholesterol (Hartmann Analytic, Braunschweig, Germany) in Dulbecco’s modified Eagle’s medium supplemented with 2% fetal bovine serum and 1% penicillin/streptomycin in the presence of 0.3 mM 8-(4-chlorophenylthio)-cyclic adenosine monophosphate (Sigma-Aldrich, Darmstadt, Germany) overnight. Cyclic adenosine monophosphate was used to upregulate ATP-binding cassette transporter A1. The day after labelling, cells were rinsed with serum-free Dulbecco’s modified Eagle’s medium containing 1% penicillin/streptomycin and equilibrated with serum-free Dulbecco’s modified Eagle’s medium containing 1% penicillin/streptomycin and 2 mg/mL bovine serum albumin (Sigma-Aldrich, Darmstadt, Germany) for 2 h. Subsequently, [^3^H]-cholesterol efflux was determined by incubating cells for 3 h with 2.8% apoB-depleted serum. Cholesterol efflux capacity was expressed as the radioactivity in the medium relative to total radioactivity in medium and cells. All steps were performed in the presence of 2 µg/mL of the acyl- coenzyme A cholesterol acyltransferase inhibitor Sandoz 58-035 (Sigma-Aldrich, Darmstadt, Germany).

### 2.16. Adipokines

Leptin, leptin receptors, and adiponectin were determined in serum samples by specific enzyme-linked immunosorbent assays in accordance with the user manual (all BioVendor, Brno, Czech Republic). In brief, samples were incubated in microplate wells pre-coated with polyclonal anti-human leptin antibody, monoclonal anti-human leptin receptor antibody, or polyclonal anti-human adiponectin antibody. After incubation and washing of recombinant human leptin, leptin receptor or adiponectin together with polyclonal anti-human leptin antibody, monoclonal anti-human leptin receptor antibody, or polyclonal anti-human adiponectin antibody conjugated with horseradish peroxidase was added to the wells and incubated with the captured analytes. After another washing step, the horseradish peroxidase conjugate bound to leptin, leptin receptor, or adiponectin immobilized on the wells reacted with the added substrate solution. By adding an acidic solution, we found that the reaction stopped, resulting in a yellowish product. The absorbance was measured photometrically, whereas the concentrations of the analytes leptin and leptin receptor were proportional and the concentration of adiponectin was inversely proportional to the absorbance. Concentrations were determined using standard curves.

### 2.17. Statistical Analysis 

Statistical analyses were performed using SPSS Statistics (version 26) and R (version 3.6.1). Differences between the overweight or obese group with the normal weight control group were analyzed using Wilcoxon rank sum test. Data are presented as medians with interquartile range. Correlations were determined using Spearman’s correlation coefficient *rho* and were Bonferroni corrected. 

## 3. Results

### 3.1. Obesity Was Associated with Alterations in HDL Composition and Subclass Distribution 

In total, 26 normal weight, 22 overweight, and 20 obese women were included in this study. Subject characteristics are presented in Table 1. 

Despite the reported key role of lipids and apolipoproteins, specifically apoA-I, in HDL metabolism, the composition of HDL particles in obesity is currently unknown. We first assessed whether overweight or obesity affected the lipid composition of HDL. Remarkably, we observed substantially lower levels of HDL-associated free cholesterol, cholesteryl-esters, and phospholipids in the obese group compared to the normal weight group (Figure 1A–C), while there was a trend towards increased HDL triglyceride content (*p* = 0.055) (Figure 1D). Interestingly, we observed no changes in the lipid composition of HDL in overweight women (BMI 25.0–29.9 kg/m^2^), except for HDL triglycerides, which were significantly elevated (*p* < 0.001) (Figure 1D).

In addition to the alterations in the lipid composition, HDL of obese women showed decreased levels of apoA-I when compared to normal weight women (Figure 2A), while the contents of apoA-II, apoC-II, apoC-III, and apoE were not altered (Figure 2B–E). We detected increased levels of the acute phase protein serum amyloid a (SAA) in HDL of obese women (Figure 2F), suggesting low-grade inflammation, in line with increased C-reactive protein (CRP) and interleukin-6 (IL-6) levels in obese women (Table 1). 

The HDL size distribution reflects the complex intravascular metabolism of these lipoproteins. On the basis of the differences in density and size, we were able to subdivide HDL particles into large and lipid-rich HDL2 particles and protein-rich smaller HDL3 particles. When assessing HDL particle distribution, we observed increased levels of the HDL3 (Figure 3A) and the small HDL3 (Figure 3B) subclasses and decreased levels of the HDL2 subclass (Figure 3C) in the obese compared to normal-weight group. A representative gradient gel electrophoresis of HDL subfractions is shown in Figure 3D.

### 3.2. Obesity Altered Activities of Enzymes Involved in HDL Metabolism

Having observed marked changes in HDL structure and composition in obese women, we next investigated whether obesity affected activities of key enzymes involved in HDL metabolism. HDL-associated cholesteryl ester can subsequently be transferred to apoB-containing lipoproteins via CETP in exchange for triglycerides. Notably, we observed increased activity of CETP in obese women when compared to the normal weight control group (*p* = 0.044), while the activity remained unchanged in the overweight group (Figure 4A). Through LCAT-induced esterification of HDL-associated free cholesterol to cholesteryl-ester, nascent HDL is converted into mature HDL, an important step in reverse cholesterol transport [40]. Interestingly, LCAT activity and protein levels were markedly increased in obese subjects when compared to the normal weight group (Figure 4B,C) but were not significantly altered in the overweight group (Figure 4B,C). LCAT activity correlated inversely with the ratio between free cholesterol and total cholesterol (Figure 4D). 

### 3.3. Functional Metrics of HDL in Overweight and Obese Women

HDL exhibits potent anti-oxidative properties [41] contributing to anti-oxidative capacity of serum. We observed a notable reduction in the ability of serum of obese women to inhibit free radical-induced oxidation of the fluorescent dye dihydrorhodamine (*p* = 0.005) (Figure 5A). This was paralleled by a marked increase in the total amount of peroxides (TOC) in serum of obese women (Figure 5B). The anti-oxidative capacity of serum correlated negatively with TOC (r_S_ = −0.58, *p* < 0.001). We observed no significant change in other metrics of HDL function, such as arylesterase activity of HDL-associated paraoxonase and HDL cholesterol efflux capacity in overweight or obese women (Figure 5C,D).

### 3.4. Correlation of Lipoprotein Parameters with Obesity-Related Factors

We observed multiple robust and complex associations of obesity-related factors with metrics of HDL structure and metabolism, depicted in a heat-map (Figure 6). Of particular interest, we observed that the HDL2 subclass inversely correlated with several liver enzymes, including alkaline phosphatase, gamma-glutamyl transpeptidase, cholinesterase, and serum alanine transaminase. Interestingly, LCAT activity showed a positive correlation with cholinesterase, as well as a positive correlation with the soluble leptin receptor. Further correlation analysis revealed that LCAT activity correlated positively with glycated hemoglobin A1c (HbA1c; r_S_ = 0.35, *p* = 0.004), HDL3 (r_S_ = 0.57, *p* < 0.001), and small HDL3 (r_S_ = 0.44, *p* < 0.001), and negatively with HDL2 subclass (r_S_ = −0.55, *p* < 0.001). Somewhat unexpected, LCAT protein levels correlated with C-reactive protein levels (r_S_ = 0.37, *p* = 0.01) and total oxidative capacity (r_S_ = 0.41, *p* < 0.001), suggesting a compensatory mechanism. CETP showed a significant correlation with hip circumference (r_S_ = 0.25, *p* = 0.44) and with D_INCL_ (r_S_ = 0.28, *p* = 0.25). Further, CETP correlated significantly with VLDL (r_S_ = 0.35, *p* = 0.004) and with plasma triglycerides (r_S_ = 0.38, *p* = 0.002).

Adiponectin levels correlated positively with HDL-C and the large HDL2 subclass, but negatively with HDL3 and small HDL3 subclasses. Levels of the soluble leptin receptor (which plays an important role in maintaining leptin sensitivity [42]) showed a correlation pattern like adiponectin. In contrast, leptin levels showed opposite correlations.

ApoC-III is a potent inhibitor of lipoprotein lipase [43] and showed a strong correlation with serum triglycerides (r_S_ = 0.59, *p* < 0.001) and HDL-associated triglycerides (r_S_ = 0.51, *p* < 0.001). As expected, we observed a positive correlation of SAA with C-reactive protein levels (r_S_ = 0.66, *p* < 0.001).

BMI and measurements of body fat, such as D_INCL_, waist circumference, and hip circumference, correlated positively with HDL3 and small HDL3, whereas large HDL2 correlated negatively and the strongest correlations were observed with the measurement of waist circumference.

## 4. Discussion

The identification of factors that modulate HDL particle distribution and subsequent HDL function is crucial for a better understanding of the complex relationship between HDL and cardiovascular risk. In the present study, we provide evidence that obesity significantly affects HDL metabolism, subclass composition, and distribution in association with changes in liver and adipose tissue. Alterations in HDL metabolism, structure, and function in obesity could contribute to increased cardiovascular risk.

In the present study, we found that obesity was associated with marked changes in CETP and LCAT activities and altered HDL composition, such as decreased apoA-I, cholesterol, and phospholipid levels, while serum amyloid a levels were increased. We observed a marked shift towards smaller HDL subclasses in obesity linked to lower anti-oxidative capacity of serum. Of note, most of these alterations were only seen in obese women but not in overweight women. We noted a strong association of the small HDL subclasses with serum triglyceride levels and an inverse association with the larger HDL2 subclass. Our results are in good agreement with a previous study showing that levels of small HDL3 particles are significantly increased in hypertriglyceridemic individuals and directly correlated with the extent of hypertriglyceridemia [44].

Of particular importance is our observation that LCAT activity and protein concentration were significantly increased in obese individuals. LCAT is reported to be closely involved in reverse cholesterol transport, an anti-atherogenic process by which excess cholesterol is removed from cells by HDL and delivered to the liver for excretion [45]. It was shown previously that LCAT activity is increased in patients suffering from type 2 diabetes and that it is supposedly connected to reduced anti-oxidative capacity of HDL [46]. Similar to this study, we observed a significant correlation of HbA1c with LCAT activity and a significant reduction in the anti-oxidative capacity of serum in obese women. Somewhat unexpected, LCAT protein levels correlated with C-reactive protein levels and total oxidative capacity. Moreover, we observed a robust positive correlation of LCAT activity with the liver function marker cholinesterase. This is somewhat surprising considering that LCAT activity was reported to be substantially decreased in patients with liver disease [35,47]. An explanation for the positive correlation of LCAT activity with cholinesterase in our study could be the fact that we determined LCAT activity in obese but otherwise young and healthy women with only slightly increased liver markers. Thus, it appears that the concentration and activity of LCAT increases with obesity and obesity-associated low-grade inflammation, suggesting a compensatory mechanism, but appears to decrease when it comes to severe comorbidities and pathological conditions [35,47,48,49].

CETP, another important enzyme involved in HDL metabolism, mediates the exchange of cholesteryl esters from HDL to triglyceride-rich lipoproteins in exchange for triglycerides [50]. In our study, we observed an increase in CETP activity in the obese group, which is consistent with the fact that adipose tissue is one of the major sources of CETP expression [51,52]. Our data suggest that higher LCAT activity in obese women may lead to increased formation of cholesteryl esters in HDL and subsequent CETP-mediated transfer to triglyceride-rich lipoproteins in exchange for triglycerides. This increases the triglyceride content of HDL, which accelerates the hydrolysis of HDL particles by hepatic and lipoprotein lipases, promoting the formation of smaller HDL particles [53,54,55]. In good agreement with this notion, we observed a robust association of CETP activity with small HDL3 subclass and subcutaneous fat mass. Moreover, levels of the triglyceride-rich VLDL particles correlated significantly with CETP activity. It has to be noted that the esterification by LCAT proceeds more slowly than the transfer of cholesteryl esters by CETP and thus represents a rate-limiting step [53,56,57].

In agreement with previous studies, we observed a shift towards the smaller HDL3 subclass and a reduction in the larger HDL2 subclass in obese individuals [58,59]. The HDL3 subfraction showed a strong correlation with LCAT activity, which is in agreement with the preferential conversion of lipid-poor pre-β particles to HDL3 catalyzed by LCAT. Increased CETP activity and increased cholesteryl ester transfer from HDL2 to VLDL in obese women is expected to decrease HDL2-cholesterol levels.

We found that VLDL levels were elevated in obese individuals compared to the normal weight group, whereas VLDL levels were not altered in overweight women. This is of importance, given that VLDL-cholesterol in particular accounts for the increased risk of myocardial infarction [60].

It is well known that in the state of obesity, fat accumulation and low-grade inflammation causes a dysregulated production of adipokines, leading to a reduction of adiponectin. We observed that adiponectin levels were reduced only in the obese group, but not in the overweight group. Noteworthy, adiponectin directly affects HDL cholesterol levels [61] by increasing apoA-I production and hepatic ATP-binding cassette A1 (ABCA1) expression [62] and by decreasing hepatic lipase activity, which reduces catabolism of HDL2 [63]. In good agreement, we observed robust associations of adiponectin with HDL2 levels and negative correlations with HDL3. Similar results have been reported in other studies [64,65].

We observed that leptin and soluble leptin receptor levels differed significantly in overweight and obese women when compared to normal weight women. Serum levels of the soluble leptin receptor were inversely associated with serum leptin levels but positively with HDL-C and serum adiponectin levels. In good agreement, similar results were observed in healthy Japanese subjects [66].

Interestingly, HDL composition was profoundly altered in obese women. We observed reduced levels of HDL-associated apoA-I, free cholesterol, cholesteryl ester, and phospholipids, whereas SAA content was increased in obese women, suggesting low-grade inflammation. In line with this assumption, we observed a robust correlation of SAA with C-reactive protein levels. Of note, in overweight women, only the triglyceride content of HDL was increased. We observed no significant effect of obesity on paraoxonase activity, which was previously reported in children and adolescents [67,68]. Furthermore, cholesterol efflux capacity was not affected by obesity despite reduced HDL cholesterol levels. A possible explanation for this paradoxical observation is the shift towards increased HDL3 and small HDL3 particles in obese women. HDL3 and small HDL3 particles are the most efficient mediators of cholesterol efflux [69] and might compensate for lower total HDL cholesterol levels.

In good agreement with our previous study [39], we observed a robust association of adiponectin with cholesterol efflux capacity.

Interestingly, an impaired functionality of the macrophage cholesterol exporter ABCA1 was reported in hyperinsulinaemia [70]. Furthermore, a previous study showed that insulin reduces HDL-mediated cholesterol efflux by inhibiting ATP-binding cassette transporter G1 and neutral cholesteryl ester hydrolase in macrophages [71]. In the present study, we did not observe a significant effect of obesity on HDL cholesterol efflux capacity. It has to be noted that HbA1c levels were in the normal range in overweight and obese women of our study.

The anti-oxidative capacity of serum was significantly decreased in the obese group compared to the normal weight group, consistent with a previous study, showing that oxidative stress increases with BMI [72].

We acknowledge limitations to this study. Due to the complex experiments and analyses, we had to keep the number of participants rather small. Further studies in larger cohorts are needed to confirm our findings.

Strengths of our study are that we measured multiple metrics of HDL metabolism, composition, structure, and function in relatively young and healthy individuals, excluding a major contribution of gender and age-associated diseases to structural and compositional alterations of HDL.

In conclusion, we demonstrated that obesity profoundly affects HDL metabolism and leads to changes in HDL composition and a shift towards small HDL3 particles. Importantly, we observed robust associations of LCAT activity, HDL subclass distribution, and HDL-cholesterol with soluble leptin receptor and adiponectin and liver enzyme activities. Of note, these alterations, with the exception of increased HDL triglyceride content, were not seen in overweight women. Obesity associated alterations in HDL subclasses, composition, and function may increase cardiovascular risk in obesity.

## Figures and Tables

**Figure 1 biomedicines-09-00242-f001:**
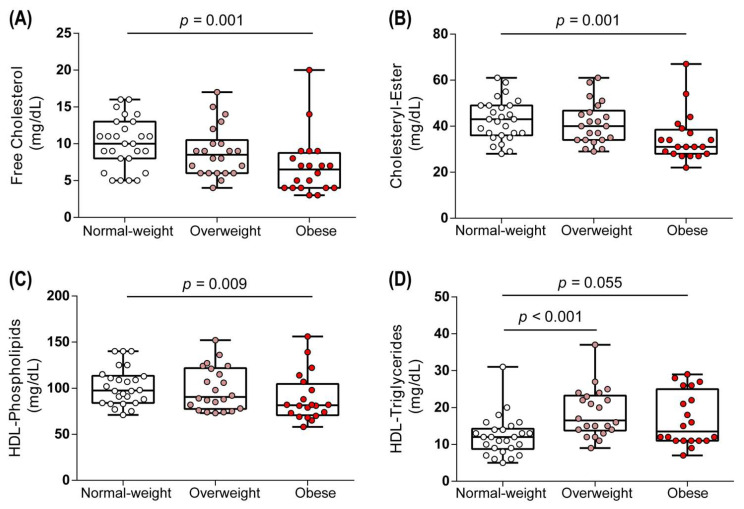
Lipid composition of HDL. Free cholesterol (**A**), cholesteryl ester (**B**), phospholipids (**C**), and triglycerides (**D**) were determined in study subjects. Differences between the two groups were analyzed by Wilcoxon rank sum test. Individual data are presented on top of boxplots displaying median and interquartile range as well as minimum and maximum values.

**Figure 2 biomedicines-09-00242-f002:**
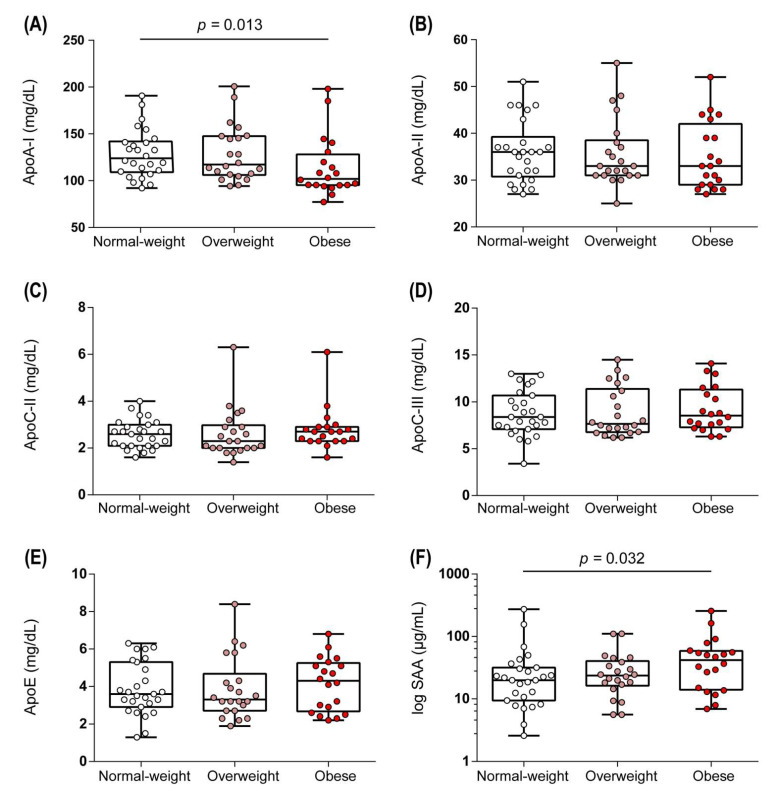
Protein composition of HDL. HDL-associated proteins, including apoA-I (**A**), apoA-II (**B**), apoC-II (**C**), apoC-III (**D**), apoE (**E**), and SAA (**F**) were determined for the study subjects. Differences between the two groups were analyzed by Wilcoxon rank sum test. Individual data are presented on top of boxplots displaying median and interquartile range as well as minimum and maximum values. ApoA-I, apolipoprotein A-I; apoA-II, apolipoprotein A-II; apoC-II, apolipoprotein C-II; apoC-III, apolipoprotein C-III; apoE, apolipoprotein E; SAA, serum amyloid A.

**Figure 3 biomedicines-09-00242-f003:**
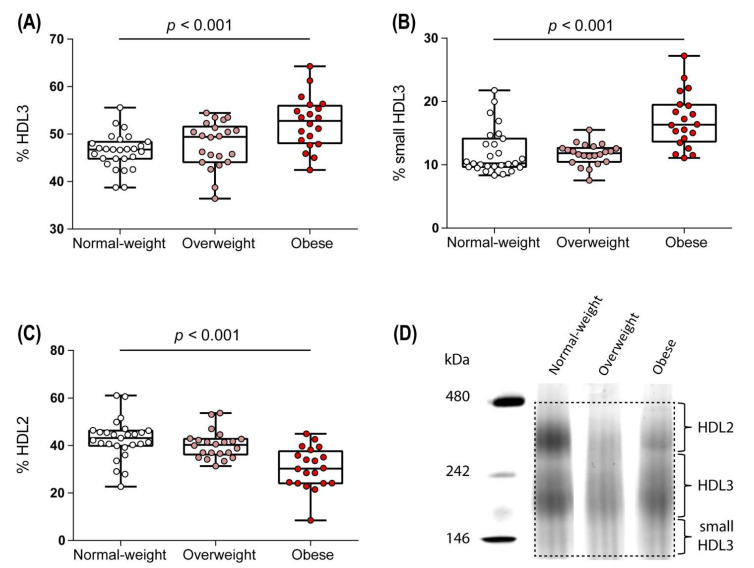
HDL subclass distribution. The distribution of HDL3 (**A**), small HDL3 (**B**), and HDL2 (**C**) subfractions was assessed by native gradient gel electrophoresis. A representative gradient gel electrophoresis of apoB-depleted serum of one normal weight, one overweight, and one obese woman is shown (**D**). Differences between the two groups were analyzed by Wilcoxon rank sum test. Individual data are presented on top of boxplots displaying median and interquartile range as well as minimum and maximum values.

**Figure 4 biomedicines-09-00242-f004:**
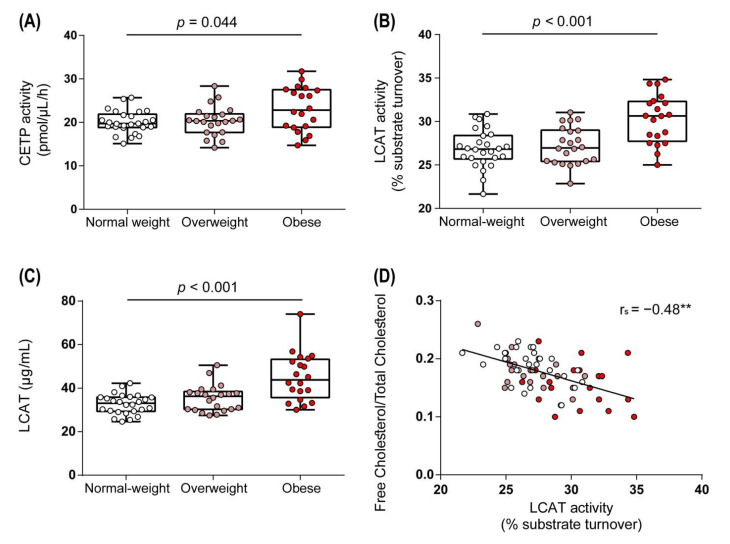
Obese women show increased serum activities of cholesteryl ester transfer protein (CETP) and lecithin–cholesterol acyltransferase (LCAT). Activities of CETP (**A**), LCAT (**B**), and LCAT protein levels (**C**) in serum samples. (**D**) shows the correlation of the ratio between free cholesterol and total cholesterol with LCAT activity. Differences between the two groups were analyzed by Wilcoxon rank sum test. Individual data are presented on top of boxplots, displaying median and interquartile range, as well as minimum and maximum values. Correlation was determined using Spearman’s correlation coefficient rho. LCAT, lecithin–cholesterol acyltransferase; CETP, cholesteryl ester transfer protein. (** *p* < 0.01).

**Figure 5 biomedicines-09-00242-f005:**
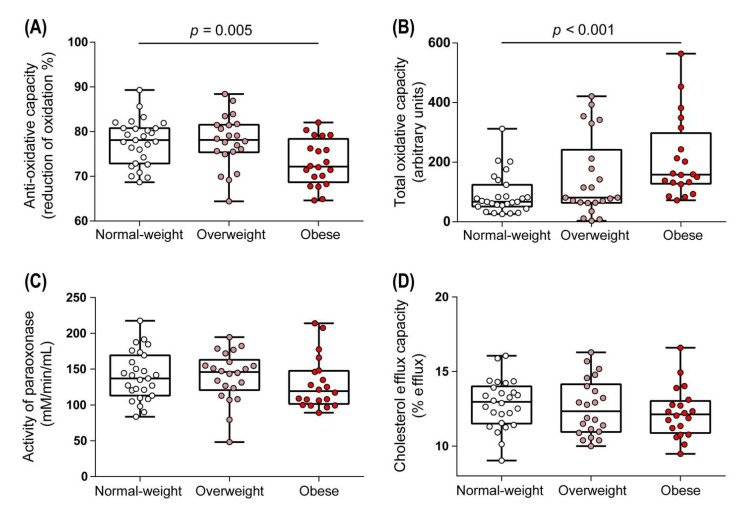
Metrics of HDL function of normal weight, overweight, and obese women. Anti-oxidative capacity of serum (**A**) and the number of total peroxides (total oxidative capacity) (**B**) were determined. ApoB-depleted sera of women were used to assess arylesterase activity of HDL-associated paraoxonase (**C**) and the ability to promote cholesterol efflux (**D**). Differences between the two groups were analyzed by Wilcoxon rank sum test. Individual data are presented on top of boxplots displaying median and interquartile range as well as minimum and maximum values.

**Figure 6 biomedicines-09-00242-f006:**
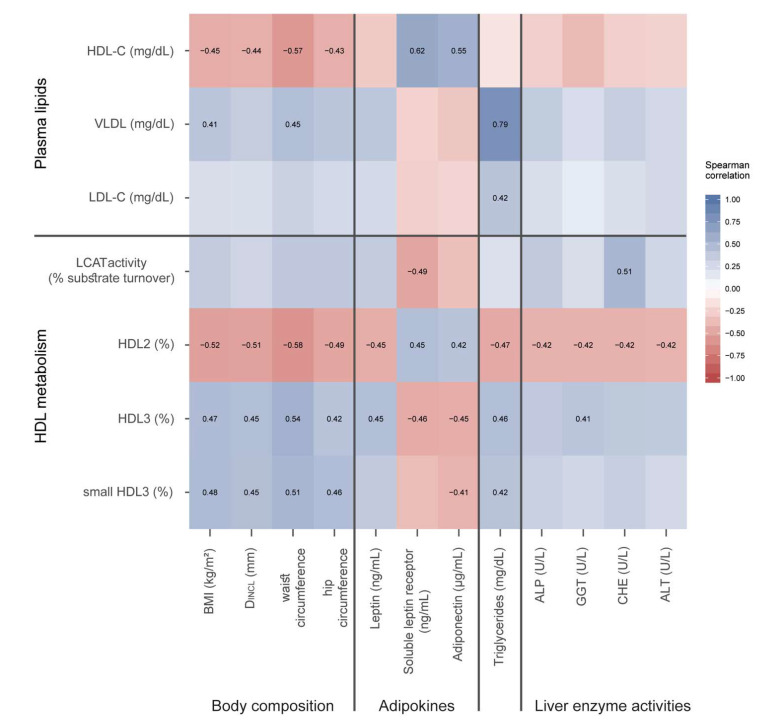
Correlation between plasma lipid levels and parameters of HDL metabolism with body composition, adipokines, serum triglycerides, and liver enzyme activities. Each cell of the heatmap represents pairwise Spearman correlation between the two parameters indicated in the respective row and column. Correlations that reached significance after Bonferroni correction are indicated with the corresponding Spearman rank correlation coefficient. Non-significant correlations can still be inferred from the color but are not explicitly indicated. HDL-C, high-density lipoprotein cholesterol; VLDL, very low density lipoprotein; LDL-C, low-density lipoprotein cholesterol; LCAT, lecithin cholesteryl acyltransferase; CETP, cholesteryl-ester transfer protein; BMI, body mass index; D_INCL_ thickness of subcutaneous adipose tissue at eight measured body sites; SOB-R, soluble leptin receptor; ALP, alkaline phosphatase; GGT, gamma-glutamyl transpeptidase; CHE, cholinesterase; ALT, alanine transaminase.

**Table 1 biomedicines-09-00242-t001:** Clinical characteristics of the study cohort.

Population Characteristics	Normal Weight (*n* = 26)	Overweight (*n* = 22)	*p*-Value	Obese (*n* = 20)	*p*-Value
Age (years)	24 (23–27)	24 (23–29)	0.976	26 (22–32)	0.373
BMI (kg/m^2^)	21.8 (20.5–23.3)	27.0 (26.3–27.4)	<0.001	33.0 (31.4–35.3)	<0.001
D_INCL_ (mm)	83.6 (66.1–99.1)	140.8 (120.2–162.0)	<0.001	196.9 (189.7–223.7)	<0.001
HDL-cholesterol (mg/dL)	80.0 (67.0–86.5)	75.0 (61.2–81.8)	0.239	58.0 (48.8–68.0)	<0.001
LDL-cholesterol (mg/dL)	80.0 (64.5–101.5)	82.0 (65.5–105.0)	0.740	106.5 (79.2–120.8)	0.031
VLDL (mg/dL)	21.0 (16.5–24.0)	22.5 (19.0–25.5)	0.309	27.5 (24.0–33.5)	<0.001
Triglycerides (mg/dL)	66.0 (50.0–88.0)	78.5 (65.5–122.8)	0.057	105.0 (79.2–143.5)	0.005
HbA1c (mmol/mol)	31.0 (30.0–32.5)	33.0 (31.0–33.0)	0.069	34.0 (32.0–36.5)	<0.001
CRP (mg/L)	1.3 (0.6–2.4)	1.4 (0.8–3.5)	0.345	5.3 (3.0–8.2)	<0.001
IL-6 (pg/mL)	1.5 (1.5–2.1)	2.0 (1.5–2.5)	0.119	3.6 (2.8–4.5)	<0.001
Leptin (ng/mL)	10.9 (7.8–14.9)	23.9 (19.1–39.2)	<0.001	49.1 (37.1–50.0)	<0.001
sOB-R (ng/mL)	19.5 (16.4–21.5)	15.1 (12.4–18.0)	0.003	10.2 (9.3–12.2)	<0.001
Adiponectin (µg/mL)	11.7 (9.4–15.5)	10.8 (9.4–12.1)	0.287	8.4 (7.0–10.5)	0.018
ALP (U/L)	50.0 (43.0–54.5)	51.0 (47.5–64.0)	0.136	60.5 (55.0–80.0)	<0.001
GGT (U/L)	12.0 (10.5–16.0)	13.0 (11.0–16.8)	0.408	17.0 (15.8–23.0)	0.002
CHE (U/L)	7225.0 (6604.0–7872.5)	7378.5 (6219.5–8535.0)	0.695	8253.5 (7682.8–9609.8)	<0.001
ALT (U/L)	14.0 (12.0–19.5)	18.0 (13.0–21.8)	0.197	23.0 (19.0–26.0)	<0.001

Clinical characteristics of the study cohort. Data are presented as median (Q1–Q3). Differences between normal weight women and either overweight or obese women were assessed by Wilcoxon rank sum test. *n*, number of subjects; BMI, body mass index; D_INCL_, thickness of subcutaneous adipose tissue at eight measured body sites; HDL, high-density lipoprotein; LDL, low-density lipoprotein; HbA1c, glycated hemoglobin A1c; CRP, C-reactive protein; IL-6, interleukin-6; sOB-R, soluble leptin receptor; ALP, alkaline phosphatase; GGT, gamma-glutamyl transpeptidase; CHE, cholinesterase; ALT, alanine transaminase.

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
