# Peer review of "Obesity Affects HDL Metabolism, Composition and Subclass Distribution"

_biomedicines, 2021, doi:10.3390/biomedicines9030242_

Round 1
Reviewer 1 Report
In this work was analyzed the effect of obesity on HDL metabolism and function. The authors analyzed different components associated to HDL, as apoliproteins, lipids and enzymes and evaluated HDL subclass distribution. Overall the work is interesting, but several aspects need to be clarified to support the conclusions:
- the authors observed a shift towards small HDL3 particles and observed a robust association of LCAT activity. How is the shape of these particles? LCAT converts the small pre-beta particles in spherical alpha migrating particles, are the HDL3 particles pre-beta or alpha migrating in obese? This aspect can clarify the paradoxical correlation between the increase of small HDL3 and the increase in LCAT activity.
- LCAT is a negative acute phase protein, that usually decrease during inflammatory state. Obese group show high levels of CRP, IL-6 and SAA. The measurement of LCAT concentration associated to results obtained by measurement of LCAT activity can help to clarify this point. Moreover, LCAT is a well know anti-oxidant enzyme and the measurement of LCAT concentration will help to clarify also this point.
- The increased CETP activity is in contrast with the reduced content of HDL2. How authors explain this discrepancy? The authors stated in discussion that it can be correlated with the accelerated hydrolysis of HDL particles by hepatic and lipoprotein lipases promoting the formation of smaller HDL particles. Since the activities of these lipases can explain different results presented in this work, the authors should measure these activities.
- The authors stated that lipids content in HDL was analyzed but the results were not shown. Figure 1 that suppose to represent HDL lipid composition reports protein composition. Figure 1 and Figure 2 are identical.
- In the results section authors stated that in obese HDL content of free and esterified cholesterol decreased. Please insert the measurement of unesterified/total cholesterol ratio to correlate with LCAT activity results.
- Figure 3D is mentioned in text but it is not reported (page 8, line 262-263)
Author Response
Reviewer 1:
In this work was analyzed the effect of obesity on HDL metabolism and function. The authors analyzed different components associated to HDL, as apoliproteins, lipids and enzymes and evaluated HDL subclass distribution. Overall the work is interesting, but several aspects need to be clarified to support the conclusions:
We are glad to hear that our manuscript was positively received by the reviewer and we are happy to consider the helpful comments.
------------------------------
The authors observed a shift towards small HDL3 particles and observed a robust association of LCAT activity. How is the shape of these particles? LCAT converts the small pre-beta particles in spherical alpha migrating particles, are the HDL3 particles pre-beta or alpha migrating in obese? This aspect can clarify the paradoxical correlation between the increase of small HDL3 and the increase in LCAT activity.
We thank the reviewer for that comment. We separated the HDL subclasses (in serum) by native gel electrophoresis. For that reason we could not stain for proteins and we had to use a lipid staining to detect the different HDL subclasses. Therefore, we were not able to detect lipid-free HDL particles or lipid-poor pre-beta particles and cannot make any prediction about the structure of the beta migrating particles. However, since besides LCAT also CETP activity was increased in the obese group, we assume that CE is effectively transferred to VLDL/LDL by CETP causing a shift from CE-rich HDL2 to HDL3.
--------------------------------
LCAT is a negative acute phase protein, that usually decreases during inflammatory state. Obese group show high levels of CRP, IL-6 and SAA. The measurement of LCAT concentration associated to results obtained by measurement of LCAT activity can help to clarify this point. Moreover, LCAT is a well know anti-oxidant enzyme and the measurement of LCAT concentration will help to clarify also this point.
According to the reviewer’s suggestion, we measured serum LCAT concentration in our study cohort. We observed similar results as for LCAT activity (see new Figure 4C). We agree with the reviewer, it is well known that LCAT activity decreases during severe inflammation. The increased LCAT activity (and LCAT protein concentrations) observed in obese women might be explained by the fact that we included overweight but young and healthy women in our study. The levels of CRP, IL-6 and SAA were elevated compared to controls, but well below what is measured in other inflammatory diseases, suggesting low grade inflammation. (page 10; lines 324-326; 386-388)
--------------------------------
The increased CETP activity is in contrast with the reduced content of HDL2. How authors explain this discrepancy? The authors stated in discussion that it can be correlated with the accelerated hydrolysis of HDL particles by hepatic and lipoprotein lipases promoting the formation of smaller HDL particles. Since the activities of these lipases can explain different results presented in this work, the authors should measure these activities.
We somewhat disagree with the reviewer on this point. All published data with CETP inhibitors clearly show that inhibition of CETP strongly increases HDL2 levels. We think it is reasonable to assume that the increased CETP activity observed in obese women will lead to a reduction in HDL2 levels. Therefore we think this is not a discrepancy.
Unfortunately, we cannot determine the activities of lipoprotein lipase and hepatic lipase in our samples. We did not administer heparin to the study participants before blood drawing, which is necessary to release lipoprotein lipase and hepatic lipase bound to endothelial cells.
--------------------------
The authors stated that lipids content in HDL was analyzed but the results were not shown. Figure 1 that suppose to represent HDL lipid composition reports protein composition. Figure 1 and Figure 2 are identical.
We sincerely apologize to the reviewers for this mistake. We now added the correct figure. (page 7)
-------------------------------------
In the results section authors stated that in obese HDL content of free and esterified cholesterol decreased. Please insert the measurement of unesterified/total cholesterol ratio to correlate with LCAT activity results.
According to the reviewer’s suggestion, we have added the correlation between free cholesterol/total cholesterol ratio with LCAT activity in Figure 4D. (page 10)
--------------------------------
Figure 3D is mentioned in text but it is not reported (page 8, line 262-263)
We thank the reviewer for pointing out this issue. Figure 3D has now been included (page 9).
Reviewer 2 Report
Stadler et al. examined in this study the effect of being overweight or obese on HDL. Essentially being overweight did not adversely affect HDL, yet being obese had effect on multiple facets of HDL including lipid and apolipoprotein content, LCAT and CEPT activities, pro-inflammatory HDL-serum amyloid, and its size and anti-oxidative capacity. This work is of interest, yet some major and minor concerns/issues with it presentation exist.
Major
Figure 1 does not present lipid composition of HDL, but rather presents data presented in figure 2.
Figure 3 does not have a panel D, indicated to show a representative gradient gel.
While the efflux capacity of the serum is not impaired, could cellular (i.e., ABCA1, ABCG1) efflux capacity be impaired? Interesting in vitro data suggest insulin resistance could be associated with impaired ABCA1 functionality; i.e., cholesterol efflux to ApoA1 (Sealls et al. Arterioscler Thromb Vasc Biol. 2011). Admittedly, this would be difficult to test. However this aspect of being overweight or obese, likely insulin resistance, could be discussed.
Minor
Lines 102-103 repeat 93-94
Author Response
Stadler et al. examined in this study the effect of being overweight or obese on HDL. Essentially being overweight did not adversely affect HDL, yet being obese had effect on multiple facets of HDL including lipid and apolipoprotein content, LCAT and CEPT activities, pro-inflammatory HDL-serum amyloid, and its size and anti-oxidative capacity. This work is of interest, yet some major and minor concerns/issues with its presentation exist.
Major
Figure 1 does not present lipid composition of HDL, but rather presents data presented in figure 2.
We sincerely apologize to the reviewers for this mistake. The correct figure has been added (page 7)
-------------------------------
Figure 3 does not have a panel D, indicated to show a representative gradient gel.
We thank the reviewer for pointing out this issue. Figure 3D has now been included (page 9)
------------------------------
While the efflux capacity of the serum is not impaired, could cellular (i.e., ABCA1, ABCG1) efflux capacity be impaired? Interesting in vitro data suggest insulin resistance could be associated with impaired ABCA1 functionality; i.e., cholesterol efflux to ApoA1 (Sealls et al. Arterioscler Thromb Vasc Biol. 2011). Admittedly, this would be difficult to test. However this aspect of being overweight or obese, likely insulin resistance, could be discussed.
Thank you for this comment. We have included the suggested study and discussed this aspect (lines 439-445)
--------------------------------
Minor
Lines 102-103 repeat 93-94
We thank the reviewer for this comment. We have deleted the repeated information
------------------------------------